# Assessment of In-Vitro Synergy of Fosfomycin with Meropenem, Amikacin and Tigecycline in Whole Genome Sequenced Extended and Pan Drug Resistant *Klebsiella Pneumoniae*: Exploring A Colistin Sparing Protocol

**DOI:** 10.3390/antibiotics11020153

**Published:** 2022-01-25

**Authors:** Manawr AL-Quraini, Meher Rizvi, Zaaima AL-Jabri, Hiba Sami, Muna AL-Muzahmi, Zakariya AL-Muharrmi, Neelam Taneja, Ibrahim AL-Busaidi, Rajeev Soman

**Affiliations:** 1Department of Microbiology and Immunology, College of Medicine and Health Sciences, Sultan Qaboos University, Muscat 123, Oman; munquraini@outlook.com (M.A.-Q.); zaeema@squ.edu.om (Z.A.-J.); m.muzahmi@gmail.com (M.A.-M.); almuharrmi@gmail.com (Z.A.-M.); 2Department of Microbiology, Jawahar Lal Nehru Medical College, AMU, Aligarh 202002, India; hibasamizafar@gmail.com; 3Department of Medical Microbiology, Post Graduate Institute of Medical Education and Research, Chandigarh 160012, India; drneelampgi@yahoo.com; 4Infectious Diseases Unit, Sultan Qaboos University Hospital, Muscat 123, Oman; ibrahimbusaidi@gmail.com; 5Jupiter Hospital Pune, Deenanath Mangeshkar Hospital Pune, Bharati University, Pune 411045, India; rajeev.soman@yahoo.com

**Keywords:** in-vitro synergy, checkerboard assay, time kill assay, fosfomycin, meropenem, *Klebsiella pneumoniae*

## Abstract

Fosfomycin has emerged as a very useful antimicrobial in management of extremely drug resistant (XDR) and pan drug resistant (PDR) *Klebsiella pneumoniae*. In this study, we assessed in-vitro synergy of colistin sparing combinations of fosfomycin (FOS) with meropenem (MEM), tigecycline (TGC) and amikacin (AK) against XDR and PDR *Klebsiella pneumoniae*. Method: Non-replicate fully characterised 18 clinical isolates of *K. pneumoniae* (15 XDR and 3 PDR strains) were subjected to in-vitro synergy testing by checkerboard and time kill assay. Combinations tested were FOS-MEM, FOS-TGC and FOS-AK with glucose-6-phosphate being incorporated in all runs.WGS was carried out on the Illumina next-generation sequencing platform. Results: FOS-MEM and FOS-AK both demonstrated excellent synergy against all PDRs and all but one XDR. Synergy led to lowering of MICs to susceptible breakpoints. FOS-TGC demonstrated antagonism. MLST-231 *K. pneumoniae* predominated (14), followed by ST-395 (3) and ST147 (1). Majority harboured OXA-232 (n = 15), while n = 2 carried NDM-1 type and n = 1 co-carried NDM-5 + OXA-232. Mortality was high in both ST-231 (57.1%) and ST-395 (66.6%). Synergy was observed despite widespread presence of resistance markers against aminoglycosides [aph(3′)-Ic, aacA4, and rmtf], beta-lactams [blaSHV-11, blaTEM-1b, blaCTX-M-15, and blaOXA-232], fosfomycin [fosA6 and fosA5] and presence of porin proteins OmpK37, OmpA and *K. pneumoniae* antibiotic efflux pumps Kpn F, H, G, and E. Conclusion: FOS + MEM and FOS + AK are excellent colistin sparing combinations against ST 231, ST-395 and ST-147 XDR and PDR *K. pneumoniae*. FOS with fewer side effects than colistin, excellent tissue distribution and minimal side effects may be recommended in combination with meropenem.

## 1. Introduction

The alarming emergence of extensively drug-resistant (XDR) and pan drug-resistant (PDR) Gram-negative bacilli (GNB) threatens the foundations of the healthcare system. XDR is defined as nonsusceptibility to at least one agent in all but two or fewer antimicrobial categories (i.e., bacterial isolates remain susceptible to only one or two antimicrobial categories) and PDR is nonsusceptibility to all agents in all antimicrobial categories [1]. XDR and PDR *Klebsiella pneumoniae* are becoming an increasing problem in Oman. Mutations in outer membrane proteins and production of an array of carbapenemases are primary drivers of XDR GNB [2]. NDM-1 and OXA-48 like enzymes in *K. pneumoniae* are increasing rapidly in the Middle Eastern region [3,4]. 

The diminishing antimicrobial arsenal for the treatment of extremely drug resistant (XDR) and pan drug resistant (PDR) Gram negative bacterial infections is a grave public health concern. One has to resort to drugs of last resort like fosfomycin, colistin, tigecycline and minocycline while treating severely ill patients. However, monotherapy with these antimicrobials may lead to development of resistance and often treatment failure. Combination antimicrobial therapies; hence, they are invaluable in the management of seriously ill patients infected with XDR and PDR Gram negative bacilli. However, selecting the appropriate combination therapy for management of these patients is a challenging task in the clinical setting. In this situation in-vitro antimicrobial synergy tests can shed valuable light on effective combinations with acceptable/minimal side effects in the management of these difficult-to-treat infections. 

For many years, colistin has been the mainstay in the management of XDR infections. However, colistin’s unpredictable pharmacokinetic/pharmacodynamic (PK/PD) interactions, poor permeability in tissues and high rates of nephrotoxicity have contributed to its limited efficacy and decreasing appeal [5].

In these drug sparse times, there is a renewed interest in an old drug-fosfomycin which is the focus of this study too. In this study, we assessed efficacy of fosfomycin as the cornerstone drug by conducting in-vitro antimicrobial synergy tests in an effort to identify effective colistin sparing combinations. 

Fosfomycin has a distinct mechanism of action as it inhibits cell wall formation by binding to the enzyme UDP-N-acetylglucosamine and thus prevents formation of the cell wall precursor N-acetylmuramic acid which is the first committed step in peptidoglycan biosynthesis. It has an extremely broad antimicrobial spectrum, being effective against XDR and PDR pathogens as well. In a study conducted by Falagas et al., fosfomycin emerged far superior to colistin, tigecycline and imipenem in treating extensively drug-resistant members of the *Enterobacterales* family [6]. Importantly, it does not share structural similarities with other antimicrobial agents, and thus lacks cross-resistance with them.

Fosfomycin’s distinct mechanism of action, its unique characteristics of high plasma concentration and tissue penetration, low cross-resistance [6] and no associated nephrotoxicity [7] makes it an attractive cornerstone drug in combination with other antibiotics. All these factors led us to hypothesize that combinations of fosfomycin with other classes of antibiotics would lead to synergy.

In this study we evaluated in vitro synergistic interactions of combinations of fosfomycin with meropenem, tigecycline and amikacin against XDR and PDR *K. pneumoniae* by checkerboard and time-kill assays and whether synergy could translate to lowering the MICs of the latter three to relevant clinical cut-offs. These drugs were selected because mechanisms of action of meropenem (inhibits peptidoglycan synthesis by binding to penicillin binding proteins), amikacin (inhibits protein synthesis) and tigecycline (inhibits protein translation) differ from that of fosfomycin thus making synergy more likely. These isolates were whole genome sequenced to identify MLST types and to understand the spectrum of antimicrobial resistance harboured by them. Synergy or lack of it was assessed in relation to this additional information. While several studies have studied synergistic activity of fosfomycin with other antimicrobials, few have assessed their efficacy in whole genome sequenced XDR and PDR *Klebsiella pneumoniae.*

## 2. Material and Methods

This study was conducted in the Department of Microbiology and Immunology, Sultan Qaboos University in collaboration with Clinical Microbiology and Immunology Laboratory and Department of Medicine, Sultan Qaboos University Hospital (SQUH), Muscat, Sultanate of Oman. Ethical approval was obtained from the Medical Research Ethics Committee, College of Medicine & Health Sciences, Sultan Qaboos University prior to commencement of study.

Eighteen (15 XDR and 3 PDR) non-duplicate consecutive genotyped strains of *Klebsiella pneumoniae* recovered from urinary tract (n = 8), respiratory tract (n = 6), wound (n = 3) and bloodstream (n = 1) infections were assessed for synergy and molecular determinants of resistance. They were identified by MALDI-TOF (Bruker Daltonics, Bremen, Germany) and antimicrobial susceptibility was carried out by BD Phoenix automated system (Becton Dickinson Diagnostic Systems, Sparks, MD, USA) as per Clinical and Laboratory Standards Institute (CLSI) guidelines [8]. The following antibiotics were tested: imipenem (IMI), meropenem (MEM), ertapenem (ETP), cefuxime (CXM), ceftriaxone (CRO), ceftazidime (CAZ), cefepime (FEP), aztreonam (ATM), amoxicillin-clavulanic acid (AMC), piperacillin-tazobactam (PTZ), gentamicin (GM), amikacin (AK), ciprofloxacin (CIP), levofloxacin (LVX), tetracycline (TET), colistin (CL), tigecycline (TGC), and trimethoprim-sulfamethoxazole (SXT). *E. coli* ATCC 25922 and *Pseudomonas aeruginosa* ATCC 27853 were used as quality control strains. Isolates were labelled as XDR if they were resistant to one agent in all but two or fewer antimicrobial groups and PDR if they were resistant to all the agents in all the groups.

Preliminary genotypic characterization was carried out by Cepheid Xpert Carba-R assay (Cepheid, Sunnyvale, CA, USA). The strains were cryopreserved and stored at −40 °C in sterile CryoBeads (Mast Diagnostics, Liverpool, UK) containing a cryopreservative fluid and glycerol with a hypertonic additive until further study.

### 2.1. Whole Genome Sequencing (WGS)

The genomic DNA was extracted from 18 to 24 h old cultures using a conventional phenol-chloroform method [9] and then purified by Qiagen kit (QIAquick PCR and Gel Cleanup Kit, 2018) as per the manufacturer’s instructions. Extracted DNA was sent to microbesNG (https://microbesng.uk, accessed on 21 February 2020, University of Birmingham, UK) for WGS (Illumina next-generation sequencing). The sequences were retrieved from the website and analysed using the tools provided by the Centre for Genomic Epidemiology, 2020 website https://www.genomicepidemiology.org/ accessed on 21 February 2020 [10,11,12,13]. ResFinder, MLST, PlasmidFinder and CSIPhylogeny tools were used to investigate the acquired antimicrobial resistance genes, multilocus sequence typing (MLST), plasmids and bacterial relatedness respectively. The CSIPhylogeny tool identified the variations between the obtained sequence data (FASTA files) by identifying and filtering high-quality SNPs (z-score higher than 1.96 for all SNPs). In addition, the CARD bioinformatics, 2020 website was utilized to look for the resistance genes and their mechanisms of action [14].

### 2.2. Minimal Inhibitory Concentrations (MIC)

MICs of fosfomycin (FOS), amikacin (AK) powders (Sigma-Aldrich Chemical Co., Saint Louis, MO, USA), meropenem (MEM, United States Pharmacopoeia) and tigecycline (TGC, European Pharmacopoeia) were determined by standard broth microdilution method using cation-adjusted Mueller-Hinton broth in accordance with CLSI guidelines [15]. For testing fosfomycin, media was supplemented with glucose-6-phosphate. ATCC 25922 (*Escherichia coli*), ATCC 27853 (*Pseudomonas aeruginosa*) were run along with each test.

### 2.3. Antimicrobial Synergy Testing

Synergy testing was performed with FOS-MEM, FOS-TGC and FOS-AK by checkerboard assay. Representative strains were tested by time-kill assay for as per Rizvi et al. [16].

#### 2.3.1. Checkerboard Assay

Broth microdilution checkerboard (BMC) was performed in 96-well microtiter plates. All tests were done in duplicate. The MICs of the other four antimicrobials weren’t affected by the presence of G-6-P. The concentrations tested ranged from ≤1/32 × MIC to 1 × MIC of each antibiotic. The interactions between antimicrobial agents were determined by calculating Fractional Inhibitory Concentration Index (FICI) wherein FICI = FIC A (MIC of drug A in combination/MIC of drug A alone) + FIC B (MIC of drug B in combination/MIC of drug B alone).The FICIs were interpreted as follows: FICI ≤ 0.5 = synergy; FICI > 0.5 to ≤ 1 = additivity (partial synergism); FICI > 1 to ≤ 4 = no interaction (indifference); FICI > 4 = antagonism [17]. 

#### 2.3.2. Time-Kill Assay

Time-kill analysis (TKA) was carried out in representative strains to confirm synergistic or additive responses obtained by the checkerboard method as previously described [18]. The bacterial cell counts for the growth control, individual antibiotic and the antimicrobial combination were plotted over time to create the time-kill curves. Synergism in TKA was defined as a 2-log10 CFU/mL decline in bacterial growth by the combination, indifference was defined as a <2-log10 increase or decrease in colony count while antagonism was defined as a 2-log10 CFU/mL increase in bacterial growth by the combination compared with the most active single agent [16]. A 3-log10 CFU/mL decrease in bacterial counts in antimicrobial combination compared with counts in the growth control indicated an adequate bactericidal response [15].

### 2.4. Statistical Analysis

The data was analyzed using IBM Statistical Package for the Social Sciences (SPSS) version 27. Paired sample *t*-test was used to test the significance of MIC reduction of the subjects (isolates) in the successful combination treatments by checkerboard assay (alone vs. combined). In all tests, differences were considered to be statistically significant when the *p*-value was <0.05.

## 3. Results

### 3.1. Clinical Profile

On analysis of the antimicrobials prescribed, eight patients (44.4%) had received combination therapy. The common combinations were MEM and CL (n = 2 cases), FOS and MEM (n = 1 cases) and triple combinations with FOS, MEM and CL (n = 3 cases). 4/6 patients receiving fosfomycin either as monotherapy (2), dual therapy (n = 1) and triple therapy (n = 1) survived as shown in Table 1. Fosfomycin monotherapy was given in two patients (one NDM-1, one OXA-48 like) and both survived. Of the seven patients who died, 2 received dual therapy of MEM and CL, 2 received triple combination of CL, MEM and FOS and three received monotherapies.

### 3.2. Distribution of Resistance Genes

Three distinct clusters were observed on analysing the concatenated alignment of single nucleotide polymorphisms using the *CSIPhylogeny* online tool https://bioinformaticshome.com/tools/descriptions/CSI_Phylogeny.html (accessed on 6 March 2020). MLST-231 (14 isolates) was the predominant cluster, followed by ST-395 (3 isolates). OXA-232 was carried on *p*KP3-A (7605 bp) replicon type ColK(P_3_) which was expressed only in the XDR OXA-232 producing *K. pneumoniae* isolates. All isolates co-harboured a large battery of resistance genes beside the carbapenemases OXA-232, OXA-48 like, OXA-1, NDM-1, NDM-5, extended spectrum beta lactamases (ESBLs) *bla_CTX-M-15_*, *bla_SHV-1,_ bla_TEM-1B,_ E. coli ampH* (*Class C β-lactamases*), OmpK37, *marA* and Kpn F, H, G, and E as seen in Table 2. OmpK37 is characterized by a narrower porin size. OmpK35 or OmpK36 were not detected. ST-395 isolates differed from ST-231 in harbouring OXA-1in addition to the other ESBLs and carried SHV-1 instead of SHV-11. The lone high-risk clone ST-147 co-harboured NDM-5 and OXA-232. Aminoglycoside resistance was mediated by *aac(3)-IId, aac(6′)-Ib and aadA2*, *aph(3′)-Ic, aacA4 and rmtf.* The PDR isolates were all ST-231clonal types, carried OXA-232 and had a similar profile of resistance genes as several XDR isolates. The plasmid-mediated fosfomycin resistance gene (*fosA*) and the mutated transporter protein *uhpT* gene were identified in all isolates. *FosA6* (fosfomycin thiol transferase) was present in ten isolates, while *FosA5* was present in ST-147 clone. Mutation in hexose-6-phosphate transport system gene (*uhpT*) results in reduced expression of uhpT, thus reducing fosfomycin uptake. Mobile colistin resistance gene (*mcr-1*) and insertion mutations in genes encoding the two-component systems *PhoPQ, PmrAB*, or in *mgrB* were not detected in any isolates.

### 3.3. Mortality and Resistance Determinants

In our study, mortality was high in both ST-231 (57.1%) and ST-395 (66.6%) infected patients. Mortality was higher (54.54%) in patients infected with OXA-232 compared with patients infected with no mortality in OXA-48 like isolates. NDM-1 infected patients had 33.3% mortality. All the PDRs carried OXA 232, MLST 231, two of whom expired. The high-risk clone ST-147 was observed in only one isolate and carried dual carbapenemases *bla* OXA-232 and *bla* NDM-5.

### 3.4. Minimal Inhibitory Concentrations

All isolates were resistant to MEM (MIC ranged from 8–128 mg/L) and TGC (2 to ≥4 mg/L) while 72.2% were resistant to AK, MIC _90_ >512 mg/L. MEM MICs (8–16 mg/L) of OXA-232 positive isolates were lower than OXA-232 + NDM-5, ST-147 isolate (128 mg/L) and NDM-1/NDM-5 producers (64–128 mg/L). The majority of isolates (13, 72.2%) were susceptible to CL (0.5–2 mg/L). High CL MICs (8 to > 64 mg/L) were seen only in the three PDRs (OXA-232) and 2 XDRs (NDM-1 and OXA-232) (Table 2). The highest sensitivity was observed in fosfomycin (88.8%, MIC: 16 to ≥64 mg/L). 

### 3.5. Synergy Outcomes 

#### 3.5.1. Fosfomycin-Meropenem Combination Outcome by Checkerboard Assay

Synergy with FOS-MEM combination was observed in all but one OXA-232 and in all three NDM-1 producing *K. pneumoniae* isolates, and across all three clonal types (MLST-231, MLST-395, MLST-147). (Table 3) Excellent synergy was observed at 1/8 MIC FOS + 1/4 MIC MEM and 1/4 MIC FOS + 1/8-1/256 MIC MEM. Synergy brought the MEM MIC down to a low MIC of 1–2 mg/L in 7/11 OXA-232, 2/4 in OXA-48 like and 1/3 in NDM-1 producing *K. pneumonia.* In isolates with higher initial MEM MIC (128 mg/L), despite synergy, MEM MIC remained at ≥16 mg/L.

One OXA-232 isolate with higher MEM MIC (128 mg/L) resulted in partial synergy (FICI ≤ 0.51) that reduced MEM MIC to clinically relevant MIC 1 and 4 mg/L, respectively, when combined with 32 mg/L FOS (individual MIC of FOS in each was 64). The significant decline in MIC suggests that partial synergy may bring about clinical cure.

#### 3.5.2. Assessment of Fosfomycin- Meropenem Combination Outcome by Time Kill Assay

On assessing FOS-MEM by TKA in the representative strain, the combination yielded a ≥3 log_10_ reduction in the bacterial population after 24 h incubation with a significant decline in the MEM MIC: from 64 to ≤1 mg/L in combination with 0.5 MIC FOS. Combination of 1/4 MIC FOS and 1/32 MIC MEM (2 mg/L) demonstrated synergy and bactericidal activity starting at 4 h and continuing for 24 h (Figure 1 and Figure 2). 

### 3.6. Fosfomycin-Amikacin Combination Outcome by Checkerboard and Time Kill Assay

A majority of AK resistant (>512 mg/L) isolates (72.2%) displayed synergy at extremely low AK MIC ≤ 0.004 (≤4–8 mg/L) (Table 4, Figure 2 and Figure 3). A striking decline in amikacin MIC from ≥512 mg/L to ≤ 8 mg/L (≥128-fold decline; *Z* = 0.001) at ≥32 mg/L fosfomycin was observed. All the ST-395 positive isolates carrying *aac(3)-IId, aac(6′)-Ib* and *aadA2* were sensitive to AK and displayed antagonistic interactions with FOS. Antagonism was displayed by the AK sensitive isolates, all three ST-395 and one each of NDM-1 and OXA-48 isolates. On assessing FOS-AK interactions by TKA, there was significant synergistic and bactericidal effect with the AK MIC of the isolate declining from >512 to ≤4 mg/L (Table 4). There was a zero count at all time intervals (Figure 3). 

### 3.7. Fosfomycin-Tigecycline Combination Outcome by Checkerboard and Time Kill Assay

No synergy was observed with FOS-TGC combination across all three MLST clones as seen in Table 5. More than two-thirds of the *K. pneumoniae* isolates (14/18; 77.8%) showed partial synergistic interactions at ≤1/8 MIC FOS + ½ MIC TGC (≥1 mg/L). Antagonism (FICI ≥ 4) was observed at 0.25–1.0 MIC FOS and a high TGC concentration (≥16 mg/L).

### 3.8. Synergy in PDR Strains

PDR strains demonstrated excellent synergy with both FOS-MEM and FOS-AK combinations with MEM and AK MIC being reduced to levels (≤2–4 mg/L and ≤16mg/L respectively). In the latter combination, FICI was significantly reduced to 0.25–0.37 (Table 5) while it was reduced to 0.50 in the former. This is noteworthy as the individual AK MIC were ≥1024 mg/L (Table 4). Antagonism and indifference were observed among the PDRs with FOS-TGC. (Table 5).

## 4. Discussion

Combination of meropenem and colistin remained the commonly prescribed combination in a large number of patients in this study, followed by a combination of fosfomycin and meropenem. It was observed that half the patients who received combination therapy recovered. It is important to note that an equal number treated by a single drug also survived.

In this study, excellent synergy was observed with FOS-MEM and FOS-AK combinations against XDR and PDR *K. pneumoniae* isolates while no synergy was observed with FOS-TGC. The majority of isolates in this study belonged to multilocus sequence type 231 (ST-231) followed by ST-395 and one was ST-147. ST-231, ST-395 co-harboured a large battery of similar resistance genes; carbapenemases (OXA-232, OXA-48 like, OXA-1, NDM-1), ESBLs (*bla_CTX-M-15_*, *bla_SHV-1,_ bla_TEM-1B,_ E. coli ampH* (*Class C β-lactamases*)), Omp K37, *marA* and Kpn F, H, G, and E carbapenems, cephalosporins, aminoglycosides and fosfomycin. ST-147 differed from the above two in carrying *Fos A5*, co-harbouring two carbapenemases OXA-232, NDM-5 and additional aminoglycoside resistance genes *armA*, *rmtB and aacA4.* Emergence of ST-147 in Oman is alarming as it has epidemic potential and as seen in this case has been reported to harbour dual carbapenemases. [19]. 

This study demonstrated excellent synergy (94.4%) between fosfomycin and meropenem against both XDR and PDR *K. pneumoniae* in all sequence types-231, 395 and 147 despite the presence of a formidable array of resistance genes. Similar synergy was also reported by Flamm et al. [20]. It is important to note that the synergy (FICI ≤ 0.50) lowered the meropenem MIC to 1–4 mg/L in 2/3rd of our isolates: 13 OXA 232/48 and one NDM-1 type. The significant reduction in MEM MIC may be because the MEM MIC (8–64 mg/L) was not very high in these isolates. FOS-MEM complement each other by targeting different stages of cell wall synthesis. TKA not only confirmed but demonstrated even higher synergy against the representative XDR NDM *K. pneumoniae* isolate. Bactericidal activity was observed as early as 6 h of incubation with 0.25 MIC fosfomycin (16 mg/L) and 1/32 MEM MIC (2 mg/L) which persisted for 24 h. Synergy levels achieved in OXA-232/48 producing *Klebsiella pneumoniae* in our study was 94% while other studies reported 20–42% synergy [21,22]. Samonis et al. reported over 70% synergy in KPC producing Klebsiella pneumoniae [23]. 

Findings of our study suggest that carbapenemase type and MIC of individual antimicrobial may determine the outcome. Synergy resulting in significant decline in MEM MIC was observed across both OXA-232 and NDM-1isolates if the initial MEM MIC ≤ 64. Partial synergy was attained if NDM positive isolates had MEM MIC ≥ 128. Animal studies are urgently needed to study if partial synergy may translate to clinical cure. 

Both FOS-MEM distribute widely in various body fluids (epithelial lining fluid of the lung, blood, urine, central nervous system) and have few side effects. MEM exhibits a time-dependent effect on bacterial killing. The pharmacokinetic/pharmacodynamic (PK/PD) index predicting its clinical and microbiological efficacy is the time (T) for which the free serum concentration exceeds the MIC [24]. The PK/PD index for MEM is T > MIC for more than 50% of dosing interval and the index for resistance suppression is T > 4xMIC for more than 50% of the time. In case of fosfomycin, the index of suppression of bacterial resistance is linked to the ratio of the area under the concentration-time curve for the free, unbound fraction of fosfomycin versus the MIC (fAUC/MIC) and T > resistance inducing concentration [25]. Administering prolonged infusions of MEM 8hrly and dosing of FOS 6hrly should be able to meet the requirements for synergy.

In this study, OXA-232 predominated. A declining trend in NDM-1 was observed which contrasts with Dortet et al.’s report in 2012 [3]. This highlights that carbapenemases are in a state of constant flux in this region. Frequent studies are needed to assess the trends, educate the healthcare givers and then develop informed treatment guidelines. 

OXA-232 is a point mutant derivative of OXA-181 with one amino acid substitution difference. In this study, OXA-232 was carried on *p*KP3-A (7605 bp) replicon type ColK(P_3_). A recent study reported that 33% of the total isolates were OXA-232 which were carried on ColKP3 plasmid (6139 bp contig) [26]. It possesses hydrolytic activities against penicillins, cephalosporins, and carbapenems that are comparable to OXA-48. Alone, this enzyme is not very potent, but with accumulation of the other resistance genes, *K. pneumoniae* acquires XDR and PDR status. In our centre the clonal group ST 231, OXA-232 emerged as the predominant variant of OXA48-like carbapenemase in Oman. Shankar et al., 2019 reported a similar trend in India [27]. 

In this study, although only one isolate was phenotypically resistant to FOS (ST 231, OXA 232, *Fos A6*, *MIC ≥ 256*), genes associated with FOS resistance were present in all of them. All isolates carried efflux pumps’ regulator gene (*marA*), mutation in the porin protein *ompk37, OmpA* plasmid-mediated fosfomycin resistance gene (fosA5, fosA6) and the mutated transporter protein *uhpT* gene. Resistance to fosfomycin is attributed to low uptake of fosfomycin because of reduced expression or chromosomal mutations in the transporter genes (*glpT*, *uhpT*), or rarely by modification in the target murA gene [28]. The latent presence of genes contributing to fosfomycin resistance suggests that combination therapy should be preferred as monomicrobial therapy with fosfomycin may rapidly select for resistant strains. Walsh et al., 2015 have also sounded the same precautionary note [29]. Efflux pumps’ regulator gene (*marA*) was detected in all isolates. Resistance to multiple antibiotics including TGC may be related to this gene as demonstrated by He et al., 2015 [30].

Fosfomycin-meropenem has been investigated and reported as an effective and synergistic combination in various studies against KPC-2, NDM, OXA-48-like, and NDM plus OXA-48-like co-producing *K. pneumoniae* in varying rates of synergy. The lowest synergy levels (20%) were observed in the study from India, although the predominant genotype was the same as in Oman-OXA-48. However, they did not report the individual strain MIC. This information would be useful to understand when to expect synergy [20]. It is interesting to note that Tseng et al., 2017 reported a synergy in 100% KPC-2 isolates with higher fosfomycin MICs compared to our study [31].

Synergism of FOS-AK in all amikacin-resistant XDR and PDR *K. pneumoniae* isolates (ST-231 and ST-147) resulted in a striking decline in AK MIC from ≥512 mg/L to ≤8 mg/L (≥128-fold decline; *Z* = 0.001) at ≥32 mg/L fosfomycin. Similar findings have been reported by Erturk et al. and Yu et al. [32,33]. The bactericidal activity of 32 mg/L fosfomycin plus 4 mg/L amikacin against the representative XDR *K. pneumoniae* (Kp7) isolate was observed as early as 2 h and persisted even at 24 h incubation. On the other hand, in the combination of 32 mg/L fosfomycin plus 1 mg/L meropenem against the representative Kp 4 isolate, bactericidal activity occurred only after 6 h incubation. Synergy yielded a statistically significant decline in the amikacin MIC (*p* = 0.001) from high MICs (≥512 mg/L), to a clinically therapeutic levels (≤8 mg/L) with ≥ a 128-fold decline at 0.5 MIC FOS (≥32 mg/L). A possible explanation for high bactericidal activity of this combination is that when two antibiotics work on different sites (e.g., cell wall and protein synthesis), they produce more effective bactericidal activity than when both work at different sites of the cell wall. However, it should be noted that isolates susceptible to AK (Kp 3, 4, 13, 14, 18) negatively interacted with fosfomycin, and the outcomes were only indifference and antagonism at various concentrations. This finding has been previously reported by us as well as by Kulengowski et al., where synergy commonly occurred in isolates characterised by high amikacin MICs [18,34]. A possible reason for this may be that synergistic concentrations are achievable in high AK MICs only.

All isolates carried resistance genes against aminoglycosides (*aac(3)-IId*, *aac(6′)-Ib* and *aadA2*, *aph(3′)-Ic*, *aacA4* and *rmtf*) and beta-lactams (*bla_SHV-11_*, *bla_TEM-1b_*, *bla_CTX-M-15_*) along with porin proteins OmpK37 and OmpA and antibiotic efflux pumps Kpn F, H, G, and E which reduce permeability to carbapenems and cephalosporin. Momin et al., 2017 reported a similar distribution of resistance genes [26]. WGS revealed that none of the three PDR isolates contained the *mcr* gene and no mutation was detected in *pmrAB*, *phoPQ*, or *mgrB* genes. Other as yet unexplored mechanisms of COL resistance must be at play.

Synergy occurred even though the ST-231 isolates carried *aac(6′)-Ib* and *aadA2* and the ST-147 isolate carried *armA*, *rmtB aadA1* and *aadA2* and all harboured *K. pneumoniae* antibiotic efflux pumps KpnF, H, G, and E. The *armA* and *rmtB* 16S rRNA methylases in ST-147 confer extreme resistance to all clinically important aminoglycosides, including gentamicin, tobramycin, and amikacin [35]. The enhanced pharmacodynamic interplay may be due to increased access of AK into target site within the bacterial ribosome after the cell wall is compromised by fosfomycin. 

TKA demonstrated that the FOS-AK combination had even greater bactericidal activity than FOS-MEM. The bactericidal activity of 32 mg/L fosfomycin plus 4 mg/L amikacin against the representative XDR *K. pneumoniae* isolate was observed at 2 h and persisted for 24 h incubation. The superiority of amikacin over carbapenems has been reported against KPC-producing and amikacin-resistant OXA-48, NDM, and OXA-48 + NDM *K. pneumoniae* isolates [20,32]. Animal studies have demonstrated that the addition of FOS resulted in protective effect against aminoglycoside-related nephrotoxicity by inhibiting aminoglycoside-induced histamine release from mast cell destruction [36]. Since AK exhibits bactericidal activity in a concentration-dependent manner, this combination will allow a higher, once-daily AMK dosing if necessary [37]. However, keeping in mind, the poor penetration of AK into lungs, bones, brain, and bloodstream, FOS-MEM should be favoured in infections related to these sites.

Surprisingly, no synergy (FICI ≤ 0.5) was observed with FOS-TGC combination, although more than two-thirds of *K. pneumoniae* isolates showed a high degree of partial synergy (FICI = 0.51). Yu et al., 2017 reported partial synergy (83.1%) and low bactericidal potential while 33% synergy was reported by Evren et al., 2013 [22,33]. Yu et al., 2017 reported antagonism in 1.5% of the isolates [33]. In a study conducted by Ku et al., 2017, synergy and bactericidal activity were reported in a third of carbapenem resistant *K. pneumoniae* isolates [38]. Several studies have recommended higher concentrations of TGC especially for the treatment of the ventilation-associated pneumonia population (VAP) [39,40]. However, if a combination of fosfomycin and tigecycline is being considered for an XDR or PDR GNB, then this study shows that a higher dose of tigecycline should be strictly avoided when combined with fosfomycin as antagonism was observed at 0.25–1 MIC FOS and a high tigecycline concentration (1 ≥16 mg/L). 

## 5. Conclusions

Synergy was observed with FOS-MEM and FOS-AK in OXA-48/OXA 232 and NDM carrying XDR and PDR *K. pneumoniae* but not with the FOS-TGC combination. This synergy was observed despite widespread presence of resistance markers against carbapenems (OXA-48/OXA 232 and NDM), aminoglycosides (aph(3′)-Ic, aacA4, and rmtf), beta-lactams (blaSHV-11, blaTEM-1b, blaCTX-M-15, and blaOXA-232), fosfomycin (fosA6 and fosA5) and presence of porin proteins OmpK37, OmpA and *K. pneumoniae* antibiotic efflux pumps KpnF, H, G, and E. FOS with enviably fewer side effects in comparison with colistin, may serve as the drug of choice in a colistin- sparing combination therapy. 

## Figures and Tables

**Figure 1 antibiotics-11-00153-f001:**
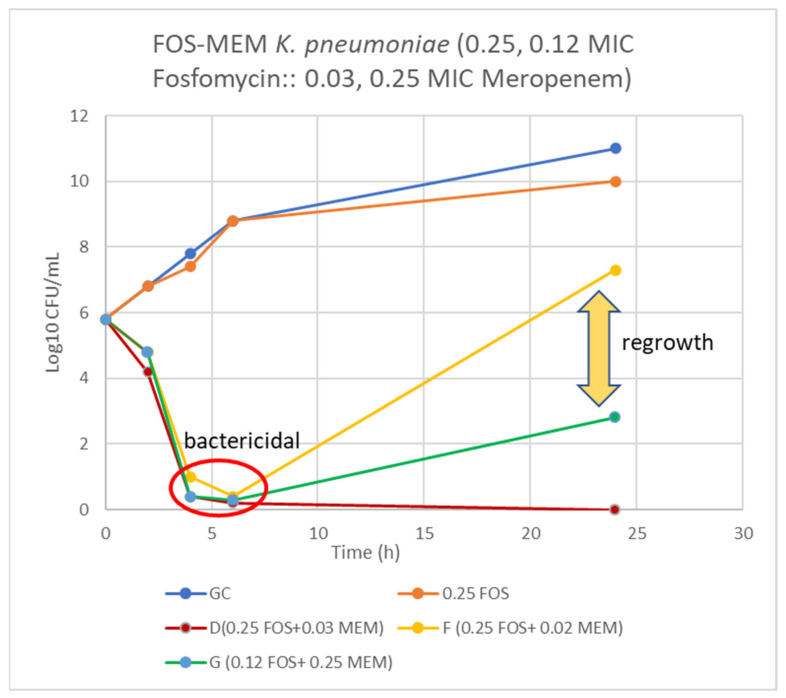
Time-kill curve of *K. pneumoniae* (Kp 4) using 0.25 and 0.12 MIC fosfomycin alone and combined with 0.03, 0.02, and 0.25 MIC meropenem. D = bactericidal effect at 6–24 h, F = bactericidal at 6 h and G = bactericidal at 4–6 h and synergy at 24 h incubation.

**Figure 2 antibiotics-11-00153-f002:**
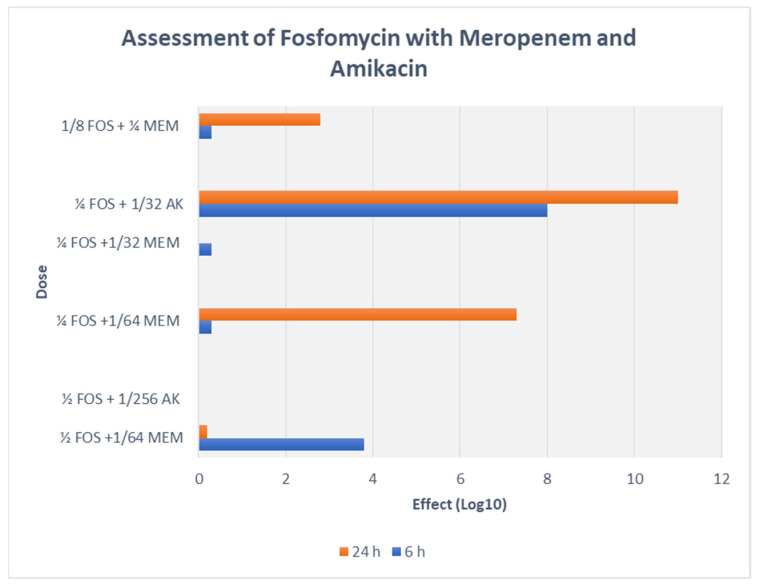
TKA assessment of FOS-MEM and FOS-AK combination with representative strain (Kp5) of *K. pneumoniae*.

**Figure 3 antibiotics-11-00153-f003:**
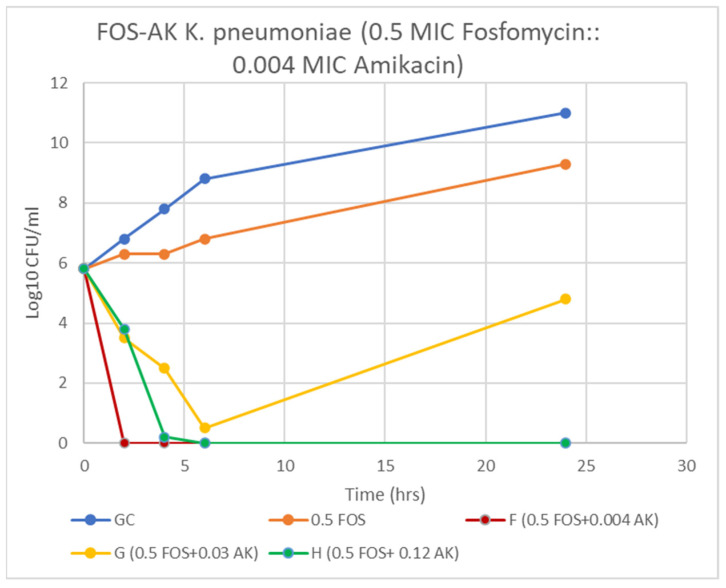
Time-kill curve of *K. pneumoniae* (Kp 5) at 0.5 MIC fosfomycin alone and combined with 4, 32, and 128 mg/L amikacin. F & H = bactericidal effect and G = a synergy at 24 h incubation.

**Table 1 antibiotics-11-00153-t001:** Demographic and clinical profiles of the patients.

Isolate	Age	Sex	Complaint	Sample	Carba-Penemase	MLST	Treatment	LOS (days)	Prior Hospital-ization	Outcome
Kp1	62	M	Acute right sided basal ganglia haemorrhage	TA	OXA-232	ST-231	Ceftriaxone 2g Q24H	134	No	Expired
Kp2	26	F	Sepsis following Chronic renal failure	TA	OXA-232	ST-231	FOS 2g Q48H + MEM 500 mg Q24H + CL 9 MU loading dose then 2MU Q12H	81	Yes	Improved
Kp3	75	F	Stroke	Urine	OXA-232		Amikacin 15 mg/kg Q24 H (900 mg Q24H)	13	No	Improved
Kp4	87	M	Pyelonephritis	Urine	NDM-1		Fosfomycin 6g Q6H (24 g per day)	10	No	Improved
Kp5	79	M	Pancreatic carcinoma Urinary incontinence	Urine	OXA-232	ST-231	None	26	No	Improved
Kp6	83	M	Aspiration pneumonia, UTI, on suprapubic catheter	Urine	OXA-232	ST-231	MEM 1g Q12H + CL 9 mU loading dose then 2.5 mU Q12H	12	Yes	Expired
Kp7	56	M	Pyelonephritis	Blood	OXA-48-like		FOS 6g 6H (total 24 g per day) + MEM 1g q8H	14	Yes	Improved
Kp8	56	M	CRE bacteremia	Urine	OXA-232	ST-231	FOS 8 q8H (total 24 g per day) + CL 9 mU loading dose followed by 4.5 mU Q12H + MEM 2g Q8H	35	Yes	Expired
Kp9	83	M	Pyelonephritis	Urine	OXA-232		FOS 6 gQ6H	0	Yes	Improved
Kp10	76	M	Bedsores	Wound	OXA-232	ST-231	None	16	Yes	Improved
Kp11	20	M	Sepsis, Chronic renal failure	TA	NDM-1		MEM 1g Q12H + CL 9 mU loading dose followed by 4mU Q8H	19	Yes	Expired
Kp12	68	M	Aspiration pneumonia	BW	OXA-232		PTZ 4.5 g Q8H + TGC 100 mg loading dose then 50 mg Q12H	12	Yes	Improved
Kp13	20	M	Gastrointestinal basidiobolomycosis	Wound	OXA-232	ST-395	CL 9 mU loading dose then 4.5 mu Q12H + MEM 2g q8H + FOS 6g Q6H (total daily dose of 24 g)	84	No	Expired
Kp14	69	M	Aspiration pneumonia	TA	OXA-232	ST-395	Tobramycin nebulization 300 mg Q12h , MEM 1g q8H + CL 9 mU loading dose then 4.5 mU Q12H	44	Yes	Improved
Kp15	49	M	Pneumonia	BAL	OXA-232	ST-231	PTZ 4.5g Q8H	10	Yes	Expired
Kp16	65	M	Metastatic gastric carcinoma	Urine	OXA-232+ NDM-5	ST-147	PTZ 4.5 g Q8H	8	No	Improved
Kp17	79	M	Acute Cystitis	Urine	OXA-48-like		Oral Fosfomycin 3g q3 days for 2 doses	-	Yes	Improved
Kp18	39	M	Haemorrhagic stroke, Bedsores	Wound	OXA-232	ST-395	PTZ 2.25 g Q8H	126	Yes	Expired

Abbreviations: LOS: Length of stay, BW: Bronchial wash, TA: tracheal aspirate, M: male, F: female, AK: amikacin, CL: colistin, MEM: meropenem, SXT: cotrimoxazole, FOS: fosfomycin, PTZ: Piperacillin-tazobactam UTI: urinary tract infection, BAL: Bronchoalveolar Lavage.

**Table 2 antibiotics-11-00153-t002:** The acquired resistance genes to different classes of antimicrobials in *K. pneumoniae* isolates.

Isolate (Kp)	MLST	Carba	MEM (MIC)	B-Lactams	TZP (MIC)	Amino	AK (MIC)	FOS	FOS (MIC)
PDR			8–16	*bla_CTX-M-15_*			>512		PDR
1			8	*bla_SHV-1_*			>512		128
2			16	*bla_TEM-1B_*		*aac(6’)-Ib aadA2*	>512		32
6	ST231	*bla_OXA-232_*	8	*E. coli ampH*	>64/4	*aph(3′)-Ic*	>512	*FosA6*	≥256
XDR			16–128	*marA,*	(R)				XDR
3		*bla_NDM-1_*	16	*Kpn F, H, G, E*			16		32
4			64	*OmpK37, OmpA*			16		64
5			16				>512		64
7			64				>512		≤32
8			64				>512		64
9			≥64				>512		64
10		*bla_NDM-1_*	64				>512		64
11			64				>512		64
15			≤32				>512		≤32
17			128				>512		64
				*bla_OXA-1_*					
XDR			8–16	*bla_CTX-M-15_*		*aac(3)-lId^*^*			
13	ST395	*bla_OXA-232_*	16	*bla_SHV-11_ bla_TEM-1B*_*	>64/4	*aac(6’)-Ib*	16	*FosA6*	32
18			8	*E. coli ampH*	(R)	*aph(3′)-Ic*	8		32
				*marA,*			(S)		
				*Kpn F, H, G, E*					
				*OmpK37, OmpA*					
				*bla_OXA-1_*					
XDR				*bla_CTX-M-15_*	>64/4	*aac(3)-IId*	8		
14	ST395	*bla_OXA-232_*	16	*bla_SHV-1_*	(R)	*aac(6’)-Ib aadA2*	(S)	*FosA6*	64
				*bla_TEM-1B_*					
				*E. coli ampH*					
				*marA,*					
				*Kpn F, H, G, E*					
				*OmpK37, OmpA*					
XDR				*bla_CTX-M-15_ bla_SHV-11_*		*armA*			
16	ST147	*bla_OXA-232_, bla_NDM-5_*	128	*bla_TEM-1B_*	>64/4	*rmtB*	>512	*FosA5*	64
				*E. coli ampH,*	(R)	*aadA1*	(R)		
				*marA,*		*aadA2*			
				*Kpn F, H, G, E*		*aacA4*			
				*OmpK37, OmpA*					

Abbreviation: PDR = pan-drug resistant, XDR = extensive drug-resistant, MLST = Multilocus sequence typing, Carba = carbapenemase, Amino = aminoglycoside, MIC = minimal inhibition concentration, FOS = fosfomycin, MEM = meropenem, AK = amikacin, TZP = piperacillin tazobactam, (R) = resistant, (S) = sensitive, * = Kp 13.

**Table 3 antibiotics-11-00153-t003:** Synergy outcomes of fosfomycin-meropenem combination using the Checkerboard assay in *K. pneumoniae* isolates.

		Fosfomycin MIC (mg/L)			Meropenem MIC (mg/L)			
Isolate	Genotype /MLST	Alone	Combined	Fold Decline	Alone	Combined	Fold Decline	FICI (x^−^)
Kp1	OXA-232/ST-231	128	16	8	8	1	8	0.25
Kp2	OXA-232/ST-231	32	4	8	16	4	4	0.37
Kp3	ST-231 OXA-232	64	16	4	16	4	4	0.50
Kp4	ST-231 NDM-1	64	8	8	64	2	32	0.16
Kp5	OXA-232/ST-231	64	8	8	16	4	4	0.37
Kp6	OXA-232/ST-231	64	8	8	8	2	4	0.37
Kp7	OXA-232/ST-231	64	8	8	16	2	8	0.25
Kp8	OXA-232/ST-231	64	8	8	16	2	8	0.25
Kp9	OXA-232/ST-231	64	16	4	16	4	4	0.50
Kp10	OXA-232/ST-231	64	16	4	8	1	8	0.37
Kp11	ST-231 NDM-1	64	8	8	128	16	8	0.25
Kp12	OXA-232/ST-231	64	16	4	8	2	4	0.50
Kp13	OXA-232/ST-395	64	8	8	16	1	16	0.19
Kp15	OXA-232/ ST-231	64	8	8	8	2	4	0.37
Kp16	OXA-232+ NDM-5/ST147	64	16	4	128	32	4	0.50
Kp17	OXA-232/ST-231	64	8	8	128	32	4	0.37
Kp18	OXA-232/ST-395	64	8	8	8	2	4	0.37

Abbreviation: Kp = *K. pneumoniae*, FICI = fractional inhibition concentration index, x^−^ = mean value.

**Table 4 antibiotics-11-00153-t004:** Synergy outcomes of fosfomycin-amikacin combination using the Checkerboard assay.

	Genotype/MLST	Fosfomycin MIC (mg/L)			Amikacin MIC (mg/L)			
Isolate		Alone	Combined	Fold Decline	Alone	Combined	Fold Decline	FICI/ Interpretation
Kp 1	OXA-232/ST-231 *aac(6′)-Ib aadA2*	128	64	2	>1024	≤16	≥128	0.50/S
Kp 2	OXA-232/ST-231 *aac(6′)-Ib aadA2*	32	16	2	>1024	≤8	≥256	0.50/S
Kp 3	OXA-232/ST-231 *aac(6′)-Ib aadA2*	64	32	2	16	128	-	>4/AN
Kp 4	ST-231 NDM-1 *aac(6′)-Ib aadA2*	64	32	2	16	128	-	>4/AN
Kp 5	OXA-232/ST-231 *aac(6′)-Ib aadA2*	64	32	2	>1024	≤8	≥256	0.50/S
Kp 6	OXA-232/ ST-231 *aac(6′)-Ib aadA2*	64	32	2	1024	≤4	≥256	0.50/S
Kp 7	OXA-232/ ST-231 *aac(6′)-Ib aadA2*	64	32	2	1024	≤4	≥256	0.50/S
Kp 8	OXA-232/ST-231 *aac(6′)-Ib aadA2*	64	32	2	1024	≤4	≥256	0.50/S
Kp 9	OXA-232/ST-231 *aac(6′)-Ib aadA2*	64	32	2	1024	≤4	≥256	0.50/S
Kp10	OXA-232/ST-231 *aac(6′)-Ib aadA2*	64	32	2	1024	≤4	≥256	0.50/S
Kp 11	ST-231 NDM-1 *aac(6′)-Ib aadA2*	64	32	2	1024	≤4	≥256	0.50/S
Kp 12	OXA-232/ST-231 *aac(6′)-Ib aadA2*	64	32	2	1024	≤4	≥256	0.50/S
Kp 13	ST-395 *aac(3)-lId aac(6′)-Ib*	64	32	2	16	64	-	>4/AN
Kp 14	ST-395 *aac(3)-IId, aac(6′)-Ib aadA2*	64	32	2	8	32	-	>4/AN
Kp 15	OXA-232/ST-231 *aac(6′)-Ib aadA2*	64	32	2	1024	≤8	≥128	0.50/S
Kp 16	OXA-232 NDM-5/ST147, *armA*, *aadA1,2,rmtB*	64	32	2	1024	≤8	≥128	0.50/S
Kp 17	OXA-232/ST-231 *aac(6′)-Ib aadA2*	64	32	2	1024	≤8	≥128	0.50/S
Kp 18	ST 395 *aac(6′)-Ib*	64	32	2	8	64	-	>4/AN

Abbreviation: FICI = fractional inhibition concentration index, Kp = *K. pneumoniae*, MIC = minimal inhibition concentration, S = synergy, AN = antagonism.

**Table 5 antibiotics-11-00153-t005:** Outcomes of fosfomycin-tigecycline combination using the Checkerboard assay.

		Fosfomycin MIC (mg/L)			Tigecycline MIC (mg/L)			
Isolate	Genotype/MLST	Alone	Combined	Fold Decline	Alone	Combined	Fold Decline	FICI (x^−^)/ Interpretation
Kp 1	OXA-232/ ST-231	128	16	≥8	2	≥32	-	>4/AN
Kp 2	OXA-232/ ST-231	32	16	≥2	4	≥32	-	>4/AN
Kp 3	ST-231 OXA-232	64	32	≥2	4	≥32	-	>4/AN
Kp 4	ST-231 NDM-1	64	≤8	≥8	4	2	2	0.62/PS
Kp 5	OXA-232/ST-231	64	≤8	≥8	4	2	2	0.62/PS
Kp 6	OXA-232/ ST-231	64	32	2	4	≥32	-	>4/AN
Kp 7	OXA-232/ST-231	64	≤8	≥8	4	2	2	0.62/PS
Kp 8	OXA-232/ST-231	64	≤8	≥8	4	2	2	0.62/PS
Kp 9	OXA-232/ST-231	64	≤8	≥8	4	2	2	0.62/PS
Kp10	OXA-232/ST-231	64	≤8	≥8	2	1	2	0.62/PS
Kp 11	ST-231 NDM-1	64	≤8	≥8	4	2	2	0.62/PS
Kp 12	OXA-232/ST-231	64	≤8	≥8	4	2	2	0.62/PS
Kp 13	OXA-232/ST-395	64	≤8	≥8	2	1	2	0.62/PS
Kp 14	OXA-232/ST-395	64	≤8	≥8	2	1	2	0.62/PS
Kp 15	OXA-232/ST-231	64	≤8	≥8	2	1	2	0.62/PS
Kp 16	OXA-232+ NDM-5/ST147	64	≤8	≥8	4	2	2	0.62/PS
Kp 17	OXA-232/ST-231	64	≤8	≥8	2	1	2	0.62/PS
Kp 18	OXA-232/ST-395	64	≤8	≥8	2	1	2	0.62/PS

Abbreviation: FICI = fractional inhibition concentration index, Kp = *K. pneumoniae*, MIC = minimal inhibition concentration, x^−^ =mean value, PS = partial synergy, AN = antagonism, OXA = Oxacillin carbapenemases, NDM = New Delhi metallo-beta-lactamase.

## Data Availability

Not applicable.

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
