# Peer review of "Assessment of In-Vitro Synergy of Fosfomycin with Meropenem, Amikacin and Tigecycline in Whole Genome Sequenced Extended and Pan Drug Resistant Klebsiella Pneumoniae: Exploring A Colistin Sparing Protocol"

_antibiotics, 2022, doi:10.3390/antibiotics11020153_

Round 1

Reviewer 1 Report

Major comments:

  1. The language of the article needs polishing.
  2. The article is illogical in some places, such as the introduction section
  3. Images need to be redone, not images with a black background.
  4. An in vivo test is necessary.

Minor comments:

  1. Line 16-17, please add n=14, n=1, n=15. Similar cases shall be modified in manuscript.
  2. Is the “Mortality was high in both ST-231 (57.1%) and ST-395 (66.6%)”your conclusion? Please provide your tests.
  3. Please deleted the first sentence in introduction in line 29-33.
  4. Please explain and define the “XDR”and “PDR”.
  5. Line 40 and 42, makes revised as make.
  6. Line 46-48, “This study aimed to evaluate in vitro synergistic interactions of fosfomycin with meropenem, tigecycline and amikacin against XDR and PDR whole genome sequenced K. pneumoniae using checkerboard and time-kill” This sentence is poor description, please modify.
  7. Line 59, Please revise “speciated”as “identified”.
  8. Line 62, please the period “. (3). ”.
  9. Line 62-65, please add the short for all drugs.
  10. L65, please add “and ”in front of “trimethoprim-sulfamethoxazole”.
  11. L65, why Pseudomonas aeruginosa ATCC 27853 as the quality control strain.
  12. L67, please add the reference.
  13. L72, “were”revise as “was”.
  14. L77, please provide the website address of CGE.
  15. L88, What is ATCC 29212 used for?
  16. L90, please delete the colon.
  17. L97, “was”revise as “were”.
  18. L109, please add the reference.
  19. L112, pleas provide the software version information.
  20. L130, provide the online tool website.
  21. L133, is OXA-1 the carbapenemase?
  22. L134, the blaCTX-M-15, SHV-1, TEM-1B don't need italics, including other section.
  23. L142, the “FosA6”represent protein, not genes,  fosA3 was gene.
  24. Table 2, what is “B-lactams”, should be beta?
  25. In table 2, are the range of MIC values of MEM 8-128 ug/mL? How do you get this value “≦32”. please add unit.
  26. L151-154, is this data from this experiment or is it quoting from other studies. Please explain. The writing of genes is wrong.
  27. L159, should be “n=13”.

Author Response

Kindly Find the attached

Reviewer 2 Report

An interesting manuscript evaluating the in vitro effect of the combination of fosfomycin with meropenem, amikacin and tigycycline on extended and pandrug-resistant Klebsiella pneumoniae strains is presented.

Due to the increasing resistance of bacterial pathogens to antibiotics, the work can be considered current and beneficial.

Evaluation of the effect of the combination of fosfomycin, an old-new antibiotic that is once again being used in the treatment of infections caused by multidrug-resistant bacteria is a suitable topic that may interest the readers of Antibiotics.

However, I have the following comments on the manuscript and I consider it appropriate that these be resolved before final acceptance.

1) I do not consider the introduction to be sufficient. There is a lack of more information on fosfomycin, its mechanism of action and the theoretical basis of synergistic activity with other antibiotics.

2) I recommend supplementing the antibiotic dosage data in Table 1.

3) Why was resistance of Klebsiella pneumoniae strains to ampicillin tested? Here is the primary resistance.

4) The discussion is too focused on the genotypic characteristics of the tested extended and pandrug-resistant Klebsiella pneumoniae strains. On the contrary, the demonstrated synergistic activity of FOS-MEM and FOS-AK is not sufficiently discussed. What are the results of further studies on this topic? How the authors explain this synergistic effect.

Author Response

Kindly Find the Attached

Round 2

Reviewer 1 Report

Some minor changes are needed, such as italics for genes and strains, and please provide a clean manuscript.

Reviewer 2 Report

Thank you to the authors for editing the manuscript according to my comments.
I believe that the article is of better quality now and I recommend its acceptance.

Author Response

Thank you so much